# Feasibility of Monitoring Tumor Response by Tracking Nanoparticle-Labelled T Cells Using X-ray Fluorescence Imaging—A Numerical Study

**DOI:** 10.3390/ijms22168736

**Published:** 2021-08-14

**Authors:** Henrik Kahl, Theresa Staufer, Christian Körnig, Oliver Schmutzler, Kai Rothkamm, Florian Grüner

**Affiliations:** 1University Medical Center Hamburg-Eppendorf, Department of Radiotherapy and Radiation Oncology, Medical Faculty, University of Hamburg, Martinistraße 52, 20246 Hamburg, Germany; henrik.kahl@stud.uke.uni-hamburg.de (H.K.); k.rothkamm@uke.de (K.R.); 2Universität Hamburg and Center for Free-Electron Laser Science (CFEL), Luruper Chaussee 149, 22761 Hamburg, Germany; theresa.staufer@desy.de (T.S.); ckoernig@mail.desy.de (C.K.); oliver.schmutzler@desy.de (O.S.)

**Keywords:** XFI, X-ray fluorescence imaging, T cell, immunotherapy, nanoparticles, gold, palladium, simulation

## Abstract

Immunotherapy has been a breakthrough in cancer treatment, yet only a subgroup of patients responds to these novel drugs. Parameters such as cytotoxic T-cell infiltration into the tumor have been proposed for the early evaluation and prediction of therapeutic response, demanded for non-invasive, sensitive and longitudinal imaging. We have evaluated the feasibility of X-ray fluorescence imaging (XFI) to track immune cells and thus monitor the immune response. For that, we have performed Monte Carlo simulations using a mouse voxel model. Spherical targets, enriched with gold or palladium fluorescence agents, were positioned within the model and imaged using a monochromatic photon beam of 53 or 85 keV. Based on our simulation results, XFI may detect as few as 730 to 2400 T cells labelled with 195 pg gold each when imaging subcutaneous tumors in mice, with a spatial resolution of 1 mm. However, the detection threshold is influenced by the depth of the tumor as surrounding tissue increases scattering and absorption, especially when utilizing palladium imaging agents with low-energy characteristic fluorescence photons. Further evaluation and conduction of in vivo animal experiments will be required to validate and advance these promising results.

## 1. Introduction

Over the past decades the field of cancer therapy has seen major breakthroughs by the implementation of novel therapeutic approaches. Most prominently, the idea of not targeting tumor cells directly but rather harnessing the body’s immune response has revolutionized the way neoplastic diseases are looked upon [1]. While this concept itself is not particularly new, it has been majorly restrained by malignant cells’ capabilities of evading immune response. This immune evasion is based on multiple mechanisms, mostly either inhibiting effector cells or disguising themselves by downregulating surface proteins [2]. Three major approaches regarding immunotherapy have been pursued, namely cancer vaccines [3], adoptive T cell transfer [4] and checkpoint inhibition [5]. The latter, working through inhibiting the activation of cytotoxic T lymphocyte-associated protein 4 (CTLA-4) or programmed cell death protein 1 (PD-1), T cell membrane proteins suppressing immune response, have shown unprecedented anti-tumor response in clinical trials [6,7], but only in a subgroup of patients. With a multitude of immune checkpoint inhibitors entering the market and multimodal therapeutic concepts being introduced, early evaluation and even prediction of treatment response is indispensable to improve the patients’ outcome [8]. Nevertheless, traditional methods of assessing tumor response mostly relying on change in tumor size were shown to be inadequate in this context due to the presence of atypical tumor responses [9]. As one of multiple markers being proposed, cytotoxic T cells (CTL) infiltrating the tumor were discovered to correlate with tumor response under these novel therapeutics and could thus be used to quickly assess and monitor the immune reaction [10]. While early investigations of T-cell distribution were limited to highly invasive biopsies not suited for routine clinical implementation [11], the emergence of molecular imaging has paved the way for non-invasive monitoring of cellular targets [12]. Four imaging modalities have been of particular interest for observing CTLs in the context of immunotherapy, either imaging injected cells or the endogenous T-cell population. The majority of the research groups apply PET and SPECT imaging due to their high sensitivity. However, as a significant efflux of radionucleotides has been observed when directly labelling T cells, the majority of PET studies has subsequently utilized radioactively labelled CD8 antibody fragments [13], with the first human trials being conducted [14]. Nevertheless, imaging of in vitro-labelled, injected T cells has also been studied using magnetic resonance imaging (MRI) [15], computed tomography (CT) [16] and bioluminescence imaging (BLI) [17]. Whereas all four approaches show promising results in either sensitivity or spatial resolution and some even allow for longitudinal imaging, none has been able to combine all aspects, with each modality having its specific drawbacks. Due to this reason, neither approach has been implemented in a clinical setting as of yet, albeit being urgently needed. Thus, we want to introduce a further imaging modality—X-ray fluorescence imaging (XFI)—and discuss the expected limits.

Ever since their discovery in 1895 [18], X-rays have been extensively studied as a way to non-invasively visualize body structures and processes. However, the possible applications of medical X-rays are far beyond the scope of established attenuation-based imaging [19]. When a photon in the X-ray energy range collides with matter, multiple interactions with the atom are possible. Regarding XFI, two effects are of major importance, namely the photoelectric effect and Compton scattering. Photons above an element-dependent energy (the so-called “absorption-edge”) have the ability to ionize atoms by removing an electron from one of their shells [20]. While the primary photon is absorbed in this process, the resulting empty position in the atom’s shell is filled by an outer shell electron, releasing energy isotropically in the form of a secondary photon. Such emissions of high energy are called X-ray fluorescence (like an “X-ray echo”) and have a characteristic energy based on the interacting atomic element. Materials with a higher atomic number Z provide higher photon emission energies [21], increasing the probability of traversing even larger targets [19]. However, detected fluorescence signal derived from these interactions is impeded by other photons, mainly originating from multiple Compton scattering, which is dependent on the target size but even applies to objects as small as a mouse. Compton scattering similarly describes the excitation of an electron through photon–atom interactions, without absorbing the primary photon but rather diverting its path. During this process, the photon loses a fraction of its energy by passing it on to an electron, depending on its energy prior to scattering and the scattering angle.

The challenge for XFI lies in differentiating the fluorescence signal from detected background photons. As multiple Compton scattering results in a shift of the incident photon energy down into the fluorescence signal region, the detection of fluorescence signals of medically feasible tracer concentrations in large targets becomes nearly impossible. However, the intrinsic problem of high Compton background has been partially overcome through the so-called spatial filtering as described in previous work by our group [22], showing the principal feasibility of XFI even for human-sized objects.

In contrast to established imaging modalities, XFI has the potential to provide both high sensitivity and excellent spatial resolution while simultaneously allowing for serial imaging to monitor targets longitudinally.

For the visualization of certain structures, specific materials with known fluorescence photon energies are used as imaging agents, commonly delivered in the form of molecular tracers or metallic nanoparticles (NPs). Nanoparticles are chosen because of their high customizability regarding size, shape and surface characteristics as well as possible functionalization, each influencing cell labelling efficiency [23,24]. NPs have been extensively studied in the context of immunotherapy, not only for imaging cellular targets [25] but also as an approach of treatment [26] and a way to specifically deliver drugs [27]. While NPs are passively accumulated in tumors due to enhanced permeability and retention (EPR) effects [28], more specific targeting is required when utilizing them in diagnostics and therapy. This can be achieved by functionalizing them through coating, change of surface structures or binding to ligands, depending on the target to be investigated [29].

Most prominently, gold nanoparticles (GNP) have been thoroughly evaluated due to their high atomic number Z (Z = 79) resulting in high X-ray absorption levels, making them well suited as a CT contrast agent [30]. GNPs have gained high popularity due to their simplicity in production and excellent customizability as well as being regarded to be of little toxicity [31]. Immune cell labelling with GNPs is usually performed ex vivo, such that only injected cells can be monitored. While cellular uptake was evaluated for a multitude of immune cells such as macrophages, monocytes and dendritic cells, T-cell labelling has rarely been performed [30]. Whereas in first studies utilizing GNPs for T-cell imaging, nanoparticles were used to deliver radionucleotides for PET imaging rather than imaging GNPs themselves [32], imaging of GNP-labelled T cells using CT was shown to be feasible [16,33]. Furthermore, the effects of GNP size, labelling duration and Au concentration on cell viability and labelling efficiency were investigated by Meir et al., achieving up to 195 pg gold per cell [23,34].

Another element that is of particular interest in cancer research is palladium nanoparticles (PdNPs). Whereas T-cell labelling has, to our knowledge, not been performed as of yet, PdNPs have been extensively researched in the context of tumor therapy, imaging and drug delivery [35]. However, there are concerns about potential toxicological and immunomodulatory effects in applications of PdNPs; hence, further investigation is needed [36]. While PdNPs have been deployed in various imaging modalities including photoacoustic imaging, SPECT and MRI are nonetheless not ideally suited for CT imaging in large objects due to the comparatively low atomic number (Z = 46) of palladium [37]. However, hybrid NPs consisting of gold-coated PdNPs have been successfully utilized in CT imaging. Regarding XFI, a reduction in tissue penetration depth is to be expected, yet it can be compensated through differing background characteristics in comparison to GNPs, as discussed in our work. Whereas gold nanoparticles have been frequently used in X-ray fluorescence studies [22,38,39], palladium is not commonly considered as an X-ray fluorescence agent.

In this work, we thus investigate the feasibility of imaging GNPs and, as a complementary approach, also PdNPs using XFI in the context of immunotherapy. For this purpose, we conducted multiple simulations using the software toolkit Geant4 v.10.5.1 [40], implementing a tumor-bearing mouse voxel model [41] irradiated by a monochromatic X-ray beam. The tumors were placed either subcutaneously, in the kidney, or in the center of the abdomen and enriched with gold or palladium nanoparticles. The software Geant4 is designed to simulate interactions of particles passing through matter using Monte Carlo methods [42,43]. Geant4-based simulations were shown to be consistent with first ex vivo and in situ experiments in previous work of our group [22,44,45] and can thus be used to explore diverse applications of XFI without the need of extensive animal research. As a result of our simulations, we intend to illuminate both prospects and challenges of implementing X-ray fluorescence in the highly topical area of cancer research to lay the foundation for future in vivo experiments.

## 2. Results

### 2.1. Subcutaneous Targets

In a first series of simulations, we mimicked imaging conditions found in several small animal imaging studies, in particular referencing a study by Meir et al. [16] utilizing GNP-labelled cytotoxic T cells for CT imaging. A tumor of 5.5 mm in diameter was placed at a subcutaneous position on the dorsolateral abdomen of the mouse voxel model, as many small animal studies investigating tumor imaging make use of artificially grown subcutaneous tumors for convenient handling. In our study, the beam enters the mouse either through the front, the back, or at a 45° angle to minimize tissue along the beam. This is achieved by rotating and repositioning the mouse, thereby influencing the amount of tissue the photons have to pass prior to and after hitting the target; however, the same effect can be achieved by repositioning of the X-ray source and detectors.

In each of the three scenarios, the significance Z was observed to be highly influenced by the detector angle. While fluorescence photons show isotropic emission, background photons predominantly composed of multiple Compton scattering are considerably influenced by the detector position. While K_α_ significance, with a signal region below the initial Compton peak, is increasingly impaired at higher detector angles through rising background, K_β_ significance does thrive with Compton energy being shifted below its signal range, as displayed in Figure 1. Moreover, the target position within the mouse has to be considered because of the influence of additional tissue between target and detector, increasing the probability of additional scattering to occur, which further reduces photon energy. When rotating the phantom to keep the tissue depth along the beam propagation axis as low as possible, the significance is further improved through an overall reduction of the Compton background. In this scenario, K_α_-fluorescence yields substantially higher significance values than K_β_ as the Compton background is less shifted towards the K_α_ region. In addition, L-shell fluorescence was examined, which will not be further discussed because of consistently performing below K-fluorescence regions. For all scenarios and signal regions, it can be observed that the significance values for gold do scale with the agent concentration, as can be seen in Figure 2. The detection thresholds for Au imaging were extrapolated to be 0.1 mg/mL for the front and back scenario and between 0.1 to 0.033 mg/mL for optimized rotation.

As can be seen in Figure 2, the Pd simulations conducted herein show superior significance across all scenarios when compared to equally enriched Au targets. However, signal intensity is substantially limited by the penetration depth of fluorescence photons emitted by Pd atoms, as can be assessed for detector angles in which the fluorescence photons have to pass the mouse’s body. When lifting these restrictions through phantom rotation, the best significance can be achieved with the detectors orthogonal (90°) to the beam direction where Compton scattering is suppressed. Derived from the dilution series, the detection threshold for Pd is estimated to be around 5 µg Pd/mL for front/back beam direction, and <3.3 µg Pd/mL for optimized rotation. The effect of Pd fluorescence being able to achieve noticeably higher significance than their Au equivalents is most prominent when the tissue thickness is as small as possible.

### 2.2. Kidney and Central Target

It was further evaluated whether this imaging technique is feasible for deep tumor imaging in small animals. This challenge was addressed by both simulating a spherical kidney lesion of 5.5 mm diameter as well as exploring the worst-case scenario, namely a similar-sized lesion in the center of the mouse with surrounding tissue of approximately 12–15 mm.

It is observed that kidney and central targets perform similarly, with the tendency of the central target position being slightly inferior. A limitation of K_α_ significance can be seen due to the shift in Compton background described before, thus lowering the maximum significance that can be achieved in these scenarios compared with previous simulations. However, when only one of both signal regions is to be examined, K_α_ remains the signal region of choice. Angular dependence is critical, with the same behavior of K_α_ and K_β_ fluorescence significance as described before. The resulting Au detection threshold for both scenarios is estimated to be between 0.33 and 0.1 mg/mL.

When utilizing Pd as an imaging agent, significance in deeper targets noticeably decreases in comparison to subcutaneous scenarios. Nevertheless, as is displayed in Figure 3, Pd does still offer higher significance than Au when comparing similar concentrations, with an observed detection limit of around 0.01 mg/mL, ergo performing at least one order of magnitude better. This is due to the different background behavior in the simulated X-ray spectra: the background in the K_α_ Pd-signal region is much less than in the K_β_-region of Au. Angular dependencies are once again mostly influenced by the thickness of surrounding tissue, performing worst for orthogonal detector positions.

### 2.3. Influence of Target Size

In further simulations utilizing the same center target scenario, the target diameter was varied between sizes of 10, 5, 2.5, and 1.25 mm. For better comparability, the simulated agent concentrations were adjusted such that the amount of fluorescence marker within the beam volume is nearly consistent for different target sizes.

The agent mass is the determining factor for the fluorescence yield; hence, the signal strength and significance Z, as scenarios differing in target size and concentration, but containing the same amount of agent mass, achieve similar results (see Figure 4). Nevertheless, it is noticed that the calculated significance does also scale with target size, albeit to a much lesser extent, as larger targets do perform slightly better than smaller ones containing the same agent mass, presumably due to the decrease of surrounding tissue. However, when looking upon the smallest target diameter, a steep decrease in significance is present, with two effects most likely contributing to this finding. On the one side, changing the shape of the beam-target intersection from cylindrical in larger targets to spherical in targets with a size approaching the beam diameter does influence the amount of fluorescence marker contributing to the observed signal. On the other hand, imperfections of beam alignment only detectable for small targets cannot be excluded.

### 2.4. Dose

Furthermore, we evaluated the deposited dose utilizing a dose tracker specified in the methods section (see Figure 5). The dose for an ideally positioned tumor is estimated to be 25.5 mGy within the tumor and 0.1 mGy of full body dose for a single beam position. When imaging a centrally placed target 5 mm in diameter, we found the organ dose to be 28.38 mGy for Au imaging and 24.45 mGy for Pd imaging with respective full body doses of 0.34 mGy (Au) and 0.33 mGy (Pd) for a single beam position. A planar scan of this 5 mm target would require 25 scan positions utilizing a 1 mm beam; thus, the full body dose for a scan is estimated to be 8.5 mGy (Au) and 8.25 mGy (Pd). As the tumor location may not always be known a priori, we also extrapolated the dose applied through a whole-body scan. When an entire mouse is scanned by a photon beam, the full-body dose roughly approaches the local dose seen within the beam volume. Applying this rationale, we estimate the full body dose to be between 250 and 350 mGy at 53 keV and 300 to 400 mGy for 85 keV.

## 3. Discussion

Based on the determined agent concentration detection limits, possible cell detection thresholds can now be estimated. As reported in the literature, 195 pg/cell Au T-cell labelling is feasible [34], whereas for Pd labelling, no reference could be found. Therefore, only the cellular detection threshold of Au is calculated, assuming a homogenous distribution of cells within the tumor. Based on our simulations of a 5.5 mm diameter tumor, the detection threshold for a subcutaneous target is estimated to be between 1.5 × 10^4^ and 4.5 × 10^4^ cells/mL (Au) for a best-case scenario and 4.5 × 10^4^ cells/mL (Au) without optimized rotation. When simulating a kidney and centered lesion, detection thresholds are extrapolated to be between 4.5 × 10^4^ and 1.5 × 10^5^ cells (Au).

When only considering the tumor volume intersecting with the 1 mm^2^ beam, it can be estimated that as little as 730 to 2400 T cells can be detected in a subcutaneous target when using gold as a fluorescence marker. For both the kidney and central target, these values increase to 2200 to 7250 cells. As shown by our simulations, the absolute mass of fluorescence marker, correlating with the number of cells within the beam, is of supreme importance. At the same time, the volume in which these cells are distributed is of minor relevance. This fact can be explained with the intrinsic strength of XFI, as the spatial resolution that can be achieved is solely limited by the beam diameter. Hence, according to our simulation study, XFI is able to detect lesions in the sub-millimeter range for scenarios in which a high cell concentration is present, with comparable spatial resolution to computed tomography and MRI, vastly surpassing the resolution achieved by PET and SPECT imaging. While this opens up promising possibilities in molecular imaging as synchrotrons are able to generate photon beams of only a few µm in diameter or even less, the size of the beam does inversely affect image acquisition times when scanning objects and more scanning positions result in increased radiation dose; thus, it cannot be shrunk ad libitum. However, when only scanning small objects and using high photon flux X-ray sources, this drawback is of minor importance.

Based on the fundamental research performed by Tumeh et al. [46], investigating the CD8+ T-cell infiltration in melanoma patients receiving immunotherapy through serial biopsies, we extrapolated the difference in T-cell abundance between responding and non-responding patients to estimate the required cell labelling efficiency needed for detection through XFI. With the data given in Figure 3 of [46], we estimate the difference of the groups to be between 1500 and 2500 CD8+ T cells per mm^2^, equaling 5.8–12.5 × 10^4^ cells/mm^3^ or 3–6.5 × 10^7^ cells in a 1 cm diameter circular tumor, assuming homogenous cell distribution. Subsequently, we calculate the required labelling efficiency to be between 0.286 and 0.044 pg Au/T cell as well as between 0.009 and 0.004 pg Pd/T cell. In a subcutaneous target, these values increase to between 0.945 and 0.133 pg Au/cell and between 0.029 and 0.013 pg Pd/cell, highlighting the potential benefit of utilizing Pd as an imaging agent due to its reduced requirements on labelling efficiency. Moreover, while these estimations only include one tumor entity and endogenous T cells; they suggest that the labelling values lie well within currently achievable Au labelling efficiencies of up to 195 pg/cell.

Furthermore, when comparing our results with established imaging modalities, the solid angle covered by the detector has to be considered. Herein, a single detector with a detection area of 25 mm^2^ (Au) or 50 mm^2^ (Pd) and a detector to target distance of 6 to 6.5 cm is used, thus only covering approximately 0.06% (Au) and 0.11% (Pd) of the full (4 π) solid angle. In comparison, modern PET, SPECT or CT detectors do feature much higher coverage, even investigating full body scanners [47]. Nevertheless, based on the observed anisotropy of the Compton background, covering the entire solid angle would not necessarily be the ideal solution for all scenarios. A detector covering 30–40% of the solid angle would be ideal for human sized targets, when targeting the K_α_ signal region of gold fluorescent agents [48], as it was previously investigated by our group.

Based on these considerations, we can assess our sensitivity data in the context of current literature regarding other imaging modalities. As mentioned above, our setup was designed such that it can provide results comparable with other studies monitoring T-cell distribution. Research performed by Meir et al. utilizing gold nanoparticles as a CT contrast agent [16] was of particular interest because the GNP-labelled cells could also be imaged using XFI. The group reported successful monitoring of injected T cell abundance in a tumor containing 8 × 10^4^ to 4.6 × 10^5^ T cells. However, no detection threshold is provided for this particular study.

Whereas this approach of injected T cells directly labelled with nanoparticles has been proven feasible, labelling endogenous cells with antibodies or antibody fragments, as has been performed in PET imaging [14], is yet to be investigated. The PET sensitivity levels that have been achieved utilizing radionucleotide-labelled anti CD8+ antibody fragments range from 2 × 10^4^ CD8+ T cells per milligram in lymphoid organs [13] to 1.6–4 × 10^6^ CD8+ lymphocytes in a tumor volume of ~480 mm^3^ [49], equaling 3.3–8.3 × 10^6^ cells/mL. Based on this data, it is to be assumed that XFI could deliver comparable detection thresholds, while simultaneously providing substantially higher spatial resolution. However, even though binding of GNPs to antibodies has been conducted in literature [50], there is a lack of studies regarding T-cell labelling using GNP-conjugated antibodies.

Direct ex vivo cell labelling of T cells has been examined in MRI studies utilizing superparamagnetic iron oxide [15,51], showing sensitivity levels of <3 labelled cells/voxel in vivo [51], with the voxel size being 75 µm × 75 µm × 500 µm, equaling <1 × 10^6^ cells/mL. However, all these publications suffered from imaging only being feasible within 48–72 h post injection, due to the fast biodegradability seen in SPIO labelling [52]. The same limitations apply for PET and SPECT imaging due to the inevitable tradeoffs made for radionuclide halftime, balancing longitudinal imaging against potential reductions in cell viability of radiosensitive lymphocytes [53]. In contrast, XFI is ideally suited for longitudinal imaging, experiencing no intrinsic decrease in signal over time. While longitudinal studies for other cell types labelled with GNPs show that imaging is feasible for at least 4 weeks [54], this number may vary for different types of cells due to variations in efflux, proliferation and cell longevity. As a fourth approach, optical imaging of T cells has achieved promising sensitivities of up to 10^4^ cells [17]; however, the translation into a clinical setting has proven difficult due to strong limitations in the tissue penetration depth as well as the complexity of cell preparation due to the required genetic cell modifications being necessary. The issue of limitations in penetration depth in XFI has been discussed in this work, endorsing previous research by showing a substantial dependance on the fluorescent agent. While palladium is limited to small animal imaging studies or the examination of superficial lesions due to its low energetic fluorescence photons, gold fluorescence in the K_α_ and K_β_ region was shown to be feasible in human-sized objects in previous investigations by our group [48,55]. However, most of these setups rely on a brilliant, monochromatic pencil beam X-ray source providing a high photon flux, features currently only achieved by synchrotron facilities, too large and too expensive for a clinical implementation. Aiming at narrowing the gap between traditional X-ray tubes and synchrotron facilities, inverse Compton X-ray sources (ICS) have been thoroughly studied over the last decades [56]. While offering quasi-monochromatic photon beams and a significantly increased photon flux in an affordable and compact size, they are often limited to lower energies [57]. To overcome these limits, our group also works on ultra-compact laser-driven Thomson X-ray sources [58], an approach envisaged for clinical use in the future.

Other approaches of achieving X-ray fluorescence imaging in a compact setup using a high-energy polychromatic X-ray tube, either as a cone or a pencil beam, have been surveyed to image tumor bearing mice [38,39,59]. In contrast to setups utilizing a pencil beam, the position of fluorescent agents in cone-beam imaging is determined using pinhole collimation, theoretically offering faster scanning times when using parallel signal attenuation in a detector array [38]. However, as such a setup that can provide a sufficient flux and narrow energy spectrum is not yet available, those methods currently suffer from the same limitations of long measurement times and a higher dose as presented in our approach. Nevertheless, great sensitivity has been achieved, even reaching synchrotron-like detection thresholds of 0.007 mg Au/mL; however, for ex vivo imaging of small targets [60], whereby using high radiation dose not suited for in vivo imaging. These drawbacks could be vastly improved in a study by Larsson et al. [39], pointing out the advantages of a pencil beam driven approach and indicating clinical applicability for molecular imaging in the sub-millimeter range for small targets.

The estimated full-body dose in the present study ranges from 0.1 to 0.34 mGy for a single beam position to 250 to 400 mGy for a whole-body scan. These doses lie markedly below the median lethal dose of 9.25 Gy after 30 days described for mice in the literature [61]. Furthermore, studies indicate that mice exposed with around 300 mGy can repair the damages within hours after exposure [62]. However, through improvements in labelling efficiency and detector size, the dose of our approach may be vastly reduced. Moreover, it should be feasible to initially locate primary and larger metastatic tumor sites by other imaging methods such as CT or MRI scans, which would then facilitate targeted XFl analysis of those specific locations.

While passive GNP attenuation in tumors has been thoroughly investigated, no study specifically targeting T cells through X-ray fluorescence has been reported as of yet, to the best of our knowledge. However, the findings presented here indicate that X-ray fluorescence imaging can be of great value in future applications of molecular imaging, specifically cell tracking. Providing similar sensitivity to established functional imaging modalities such as PET and SPECT imaging while at the same time achieving spatial resolutions usually only seen for morphological imaging in MRI and CT, XFI is ideally suited for the application of monitoring the intratumoral cell abundance. Moreover, a huge benefit over radioisotope-based approaches lies in improved longitudinal evaluation through serial imaging.

While our findings indicate a promising future for X-ray fluorescence-based imaging, its scope is a feasibility pre-study without animal research, but will be of help for test animal proposals. Hence, we believe that this pre-study paves the way for future XFI-based cell tracking research. When additional research in all areas of XFI, ranging from X-ray sources to novel labelling techniques to detector improvements, adds to the already promising capabilities of this method and allow for clinical implementation, it may play a decisive role in tumor imaging and a variety of other applications.

## 4. Materials and Methods

### 4.1. Geant4

The setup used herein consists of 3 major elements: the mouse model, detectors and additional elements modelling a beam line at PETRA3 at DESY [63]. The general setup is illustrated in Figure 6.

#### 4.1.1. Mouse Model

We utilized a segmented 3D-voxel model published and described by Dogdas et al. [41], derived through co-registration of CT and cryosection mouse data, dividing a 28 g nude male mouse into 78.4 × 10^6^ cubes. Each voxel is assigned to structures/organs with defined chemical composition and properties such as density. This material data for both the mouse voxels as well as for other objects was derived from the integrated Geant4 database. The mouse model is available free of charge and can be downloaded at the website of the Biomedical Imaging Group at the University of Southern California [64]. The model was implemented using half the maximum resolution, with 104 × 496 × 190 = 9,800,960 cubic voxels and a voxel size of 0.2 mm. The mouse position was adjusted based on the target position, such that the target always remained at the center of the setup.

#### 4.1.2. Detectors

A total of 9 detectors were added horizontally at the target height, ranging from 10 to 170 degrees with respect to the beam direction with a spacing of 20 degrees, thus covering the entire semicircle. The detector-to-target distance was 6 cm in a subcutaneous scenario and 6.5 cm in a kidney and center scenario to avoid geometric interferences between the detectors and the tube.

For the simulations targeting Pd, a GEANT4 implementation of an Amptek silicon drift diode detector was modeled (70 mm^2^ FAST SDD^®^; Amptek Inc., Bedford, MA, USA). This detector offers an active detector area of 70 mm^2^ collimated to 50 mm^2^ with a silicon sensor thickness of 500 µm, providing a superior energy resolution to planar detectors [65]. The energy resolution in the region of Pd K-shell fluorescence is on the order of 120 eV (rms). However, for higher energy X-rays, the active area made of silicon does not offer enough stopping power; thus, Amptek only recommends detector usage for energies up to 30 keV [66].

For gold simulations, a Cadmium-Telluride Detector by Amptek (XR-100CdTe; Amptek Inc., Bedford, MA, USA) was implemented with an active region of 25 mm^2^ and 1 mm thickness. CdTe detectors offer a higher quantum efficiency, with the downside of a lower energy resolution of 530 eV (rms) at 14.4 keV [67]. For both detector types, multiple detector-specific effects are considered which influence their performance [68].

#### 4.1.3. Additional Geometry

The mouse model was embedded in a plexiglass tube of 2 mm thickness and of an outer diameter of 30 mm. Moreover, a vague display of a typical beamline setup was added, consisting of an aluminum stage mounting plate. In its center, an inlay made of iron is placed to reduce scattering contributions. A stainless-steel front plate is added to block scattering of air molecules along the beam path.

### 4.2. Parameters

#### 4.2.1. Target Position

Three distinctive positions of a spherical target were simulated through specifying its center and adding agent material to each voxel containing soft tissue (density ρ = 1035 mg/mL) within a defined radius. In the case of the tumor protruding the model (subcutaneous position), an additional 0.5 mm thick layer of soft tissue was added to mimic skin.

#### 4.2.2. Agent Concentration

The agent material was chosen to be either gold or palladium. A dilution series with an agent concentration of (1, 0.33, 0.1, 0.033, 0.01, 0.0033) mg/mL inside the simulated tumor volume was performed in each scenario to cover a broad range of concentrations. For the variation of the target size, a different dilution series was chosen to improve comparability between data points (1, 0.5, 0.25, 0.125, 0.0625, 0.03125, 0.015675, 0.0078375) mg/mL, as the target diameter was also altered by a factor of 2.

#### 4.2.3. Beam

A monoenergetic photon X-ray beam of 0.5 mm radius, horizontally polarized relative to the lab frame and containing a total photon number of 10^10^, was used to keep the applied dose low while maintaining sufficient signal intensity. Optimum beam energy and detector type depend on the agent material as a tradeoff between signal intensity and background behavior, as seen in previous work of our group examining ideal beam energy [48]. In the case of Au, a beam energy of 85 keV was chosen, and for palladium we have selected 53 keV.

### 4.3. Simulation and Analysis

#### 4.3.1. Histogram Generation and Analysis

The entire spectrum is not evaluated but only predefined signal regions around the distinct fluorescence peaks. The width of these regions is defined as (E_Fluo_ − 3σ; E_Fluo_ + 3σ), with well-known fluorescence line energies E_Fluo_ specific for the target material [21] and the standard deviation σ equivalent to the energy resolution at E_Fluo_ depending on the used detector model [66,67]. Due to limitations in detector resolution as of today, certain fluorescence peaks being close together such as Pd K_α1_ and K_α2_ cannot be effectively discriminated and are therefore analyzed as one peak. 

Within these defined signal regions, the significance of a fluorescence signal can be calculated by analyzing both signal and background photons and performing a one-tailed hypothesis test. The null hypothesis H_0_ is defined as the non-existence of fluorescence photons in an observed spectrum, stating that the observed behavior is entirely explainable through background characteristics [48]. As it is also discussed in [48], the total number of photons counted when adding a fluorescent agent is not expected to be less than for Compton background alone; thus, a one-tailed test can be performed. The resulting *p*-value states the probability that an effect; herein, an increase in photon counts, is observed, albeit H_0_ being true. However, as small *p*-values are inconvenient to handle, it is converted to the significance Z, expressed as the number of standard deviations σ, stating by which probability the null hypothesis H_0_ is to be discarded. It can be approximated using:(1)$$Z\;\approx \;\frac{\left(n_{observed}-n_{expected}\right)}{\sqrt{\left(n_{expected}\right)}}\;=\;\frac{N_{signal}}{\sqrt{N_{background}}},$$
with *N_signal_* as the number of observed fluorescence photons and *N_background_* as the detected background photons [55]. In the context of this research, a significant signal is defined as Z ≥ 5σ, often considered as the significance required for discovery in particle physics [69], as previous experiments conducted by our group indicate that this value is sufficient for the detection of simulated fluorescence in experiments, indicating that even Z ≥ 3σ could be satisfactory [48,55]. Z ≥ 5σ is equivalent to a type 1 error probability of *p* ≤ 2.867 × 10^−5^%.

#### 4.3.2. Dose

A dose tracker was implemented in our simulations, storing the deposited energy for all voxels of the mouse model. Therefore, we are able to determine the dose for each organ segmented in the model as well as for each material individually.

## Figures and Tables

**Figure 1 ijms-22-08736-f001:**
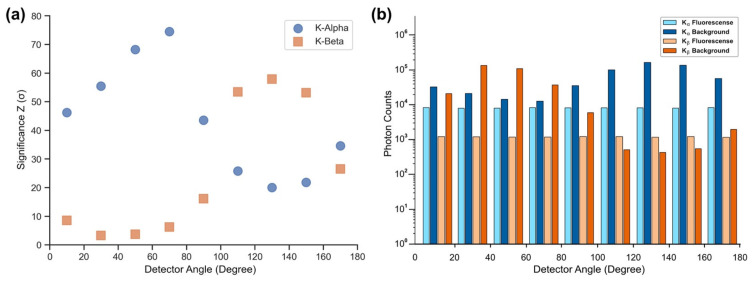
Au fluorescence significance and photon counts as simulated for an agent concentration of 1mg/mL. (**a**) Significance values for both K_α_ and K_β_ plotted for each of the 9 detector angles. (**b**) Box chart displaying the detected photon counts for both K_α_ and K_β_ fluorescence photons and background photons at the different detector angles.

**Figure 2 ijms-22-08736-f002:**
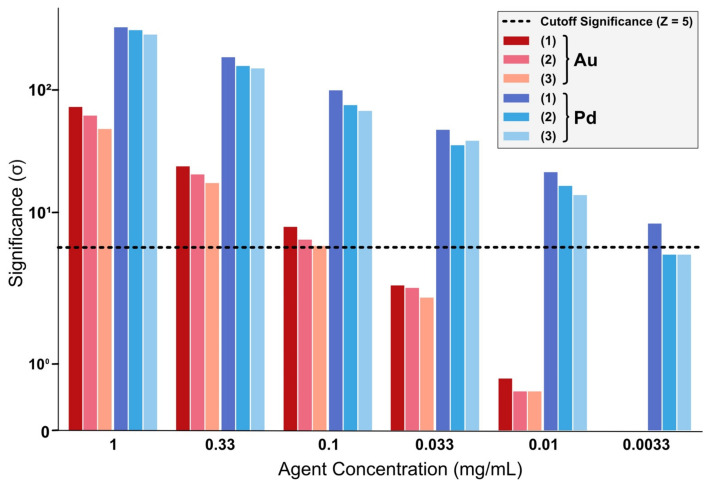
Significance comparison for a subcutaneous target at (1) optimized rotation, (2) beam hitting the mouse from the front or (3) beam entering the mouse from the back. Gold fluorescence significances are displayed in red, palladium fluorescence significances in blue.

**Figure 3 ijms-22-08736-f003:**
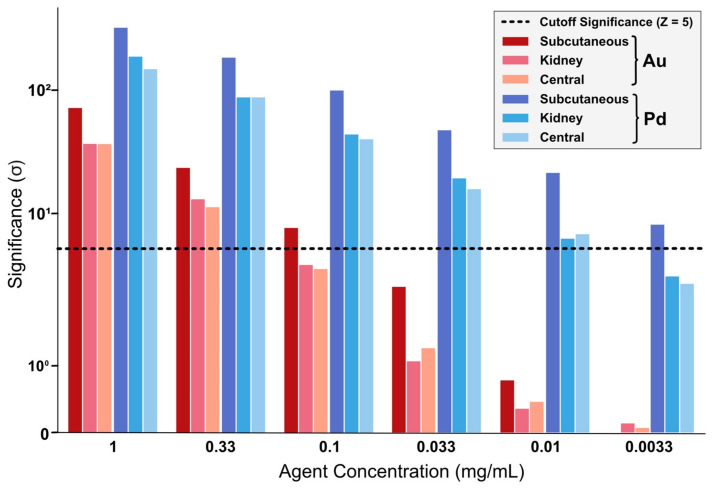
Comparison of a subcutaneous, a kidney and a centered target regarding Au-fluorescence (red) and Pd-fluorescence (blue) significances.

**Figure 4 ijms-22-08736-f004:**
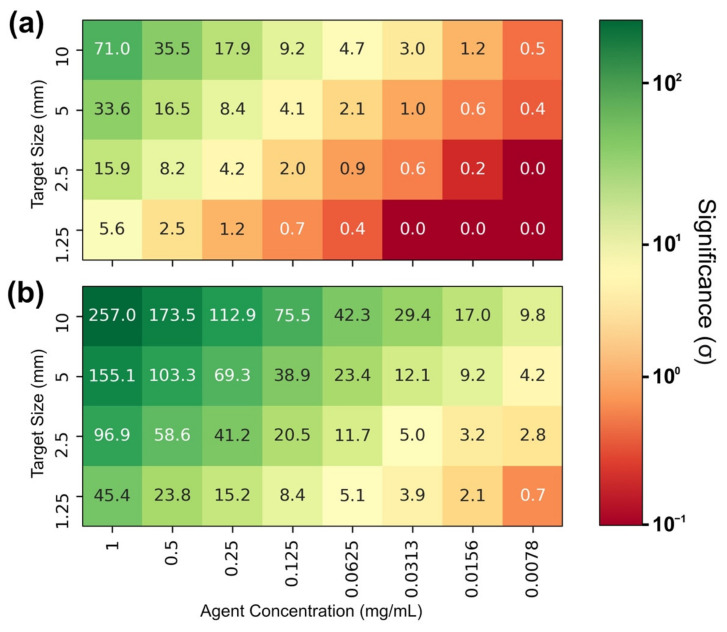
Significance Z at varying agent concentration and target size with Au-fluorescence (**a**) and Pd-fluorescence (**b**).

**Figure 5 ijms-22-08736-f005:**
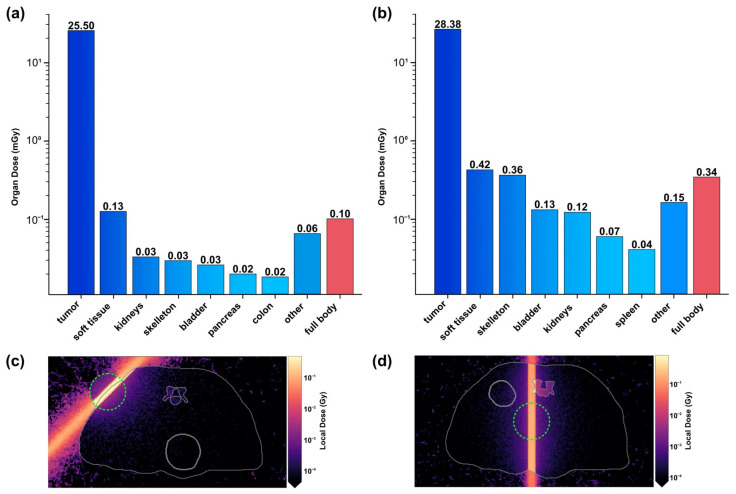
Dose deposition when imaging Au-labelled subcutaneous (**a**–**c**) and central (**b**–**d**) targets. (**a**,**b**) Organ doses and full body dose. (**c**,**d**) Transversal slice of the mouse voxel model showing the local dose at beam level. The target is highlighted by a green circle.

**Figure 6 ijms-22-08736-f006:**
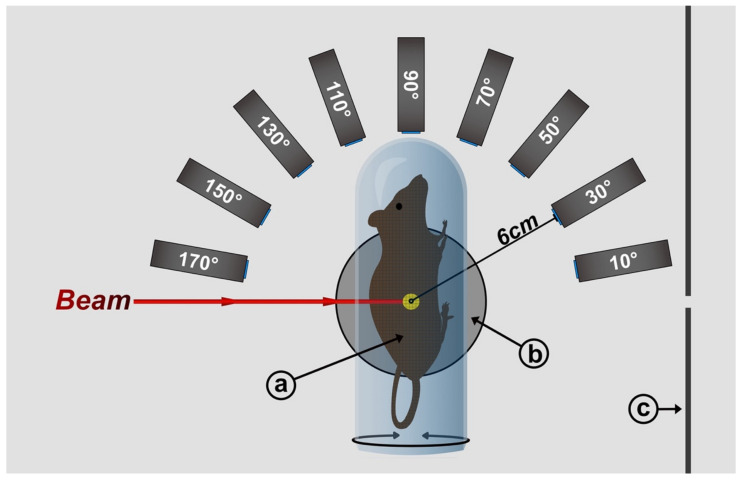
Illustration of the simulation setup as seen from above. (**a**) Rotational mouse voxel model enclosed in a plexiglass tube; (**b**) stage inlay + stage plate; (**c**) front plate.

## Data Availability

The data presented in this study are available on reasonable request.

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
