# Peer review of "Feasibility of Monitoring Tumor Response by Tracking Nanoparticle-Labelled T Cells Using X-ray Fluorescence Imaging—A Numerical Study"

_ijms, 2021, doi:10.3390/ijms22168736_

Round 1

Reviewer 1 Report

The authors evaluated the feasibility of X-ray Fluorescence Imaging (XFI) to track immune cells and thus monitor the immune response. Lacking of animal experiment to demonstrate the conclusion. I would suggest to compare different tumor models, such as subcu., i.v. and i.p. implantation, to monitor the tumor infiltrating T cells.

Author Response

We thank the reviewer for taking the time to examine our submission entitled “Feasibility of Monitoring Tumor Response by Tracking Nanoparticle Labelled T Cells Using X-Ray Fluorescence – A Numerical Study”. We highly appreciate the comments made and have addressed them point by point. We believe that we have improved the manuscript with the revisions and would be delighted to respond to any further questions or comments the referee may have.

Comment 1: Lacking of animal experiment to demonstrate the conclusion.

Response 1: We appreciate the reviewers’ suggestion of adding animal experiments to substantiate the results presented in this work. We thoroughly agree that there is a need for validation of the data presented herein through animal research in the future. Therefore, we have revised the conclusion of our manuscript to further emphasize this limitation.
However, the research design was cautiously selected to explore this topic through elaborate simulations, with the aim of primarily examining the general feasibility and evaluate if future animal experiments are expedient. As research guidelines call for replacement of animals wherever possible, we believe that our approach provides the needed foundation to evaluate the necessity of animal experiments prior to the application in novel research areas and paves the way for future experiments in this field. For this reason, we chose not to make the change, however we added the following sentence to the discussion:
While our findings indicate a promising future for X-ray fluorescence-based imaging, its scope is a  feasibility pre-study without animal research, but will be of help for test animal proposals. Hence, we believe that this pre-study paves the way for future XFI-based cell tracking research.” (Page 10-11, line 398-406)

Comment 2: I would suggest to compare different tumor models, such as subcu., i.v. and i.p. implantation, to monitor the tumor infiltrating T cells.

Response 2: We agree that different tumor positions should be examined. While we, for the reasons stated above, did not perform animal research with implanted tumors at different locations, we evaluated both subcutaneous and intraperitoneal tumor positions as well as size variations in our simulations and discussed the observed differences. Our research is focused on solid tumors only, hence i.v. injection is out of our scope.

We would like to thank the referee again for reviewing our manuscript. 

Reviewer 2 Report

The authors investigated the feasibility of X-ray Fluorescence Imaging to track immune cells using Monte Carlo simulations. The paper is written in a good style with a clear, logical path. According to the presented results, it is possible to track immune cells labeled with spherical nanoparticles of gold or palladium. While the discussed approach to detect immune cells' location and concentration is promising, there are several drawbacks the author has to answer in the discussion section.

  1. In preclinical in vivo studies, subcutaneous tumor in the dorsolateral abdomen is easy to locate and target by the beam. But studies using a metastatic model or spontaneous tumor model will require a whole-body scan. Which, taking into account low energy X-rays, will result in a high absorbed dose. Possibly lethal to the study subject. How do authors propose to resolve this issue?   
  2. It will be interesting to see the radiation pattern of Au and Pd fluorescence X-rays versus the radiation pattern of the noise photons (Compton and bremsstrahlung) for detector angles between 10 and 170 degrees.

Minor edits:

  1. Line 438. It is intuitive understandable that dilution series (1, 438 0.33, 0.1, 0.033, 0.01, 0.0033) mg/ml are set inside the simulated tumor volume. But it is not stated on paper.   
  2. At line 444, Relatively to what is the X-ray beam horizontally polarized? 

Thank you

Author Response

We thank the reviewer very much for the time and effort he has dedicated to providing the valuable feedback regarding our manuscript titled “Feasibility of Monitoring Tumor Response by Tracking Nanoparticle Labelled T Cells Using X-Ray Fluorescence – A Numerical Study”. We highly appreciate the insightful comments. We have addressed each of them below and believe, that the revised manuscript is greatly improved with the incorporated changes.

Comment 1: In preclinical in vivo studies, subcutaneous tumor in the dorsolateral abdomen is easy to locate and target by the beam. But studies using a metastatic model or spontaneous tumor model will require a whole-body scan. Which, taking into account low energy X-rays, will result in a high absorbed dose. Possibly lethal to the study subject. How do authors propose to resolve this issue?

Response 1: We agree with the reviewer, that a high absorbed dose would be a serious limitation of our approach. To address the concerns, we have modified the paragraph discussing the expected dose by adding further estimates to whole body-scans and references regarding the toxic effects of radiation in mice. We have added the sentences “As the tumor location may not always be known a priori, we also extrapolated the dose applied through a whole-body scan. When an entire mouse is scanned by a photon beam, the full-body dose roughly approaches the local dose seen within the beam volume. Applying this rationale, we estimate the full body dose to be between 250 and 350 mGy at 53 keV and 300 to 400 mGy for 85 keV.” (Page 7, line 256-261)

For a whole-body scan, the expected full-body dose will approach the local dose seen within the beam volume. Hence, we can estimate, that the full body dose for such scans would likely be between 250 and 400 mGy. With reference to the existing literature, this would most likely not induce lethal or long-lasting damage as multiple studies suggest regeneration of mice exposed with up to 300 mGy (Parkins et al., 1985), whereas the median lethal dose after 30 days (LD50/30) is described to be around 9.25 Gy (Sharma et al., 2020). In another interesting study, researchers found that 400 mGy of radiation 6h before applying a mid-lethal dose to mice decreased the survival, whereas 400 mGy at 24h before the higher dose did not, indicating repair mechanisms within that time frame (Ito et al., 2007). These new references are given below and added to the manuscript.
Moreover, with potential improvements in labelling efficiency and detector technology and size, the number of photons needed for our approach and thus the applied dose may be vastly reduced in the future. Additionally to the revisions on page 7, we have integrated the paragraph “The estimated full-body dose in the present study ranges from 0.1 to 0.34 mGy for a single beam position to 250 to 400 mGy for a whole-body scan. These doses lie markedly below the median lethal dose of 9.25 Gy after 30 days described for mice in the literature [61]. Furthermore, studies indicate that mice exposed with around 300 mGy can repair the damages within hours after exposure [62]. However, through improvements in labelling efficiency and detector size, the dose of our approach may be vastly reduced. Moreover, it should be feasible to initially locate primary and larger metastatic tumor sites by other imaging methods such as CT or MRI scans, which would then facilitate targeted XFl analysis of those specific locations” (Page 10, line 379-387)
We apologize for the previously missing information regarding whole body exposure and hope this makes it clearer.

Comment 2: It will be interesting to see the radiation pattern of Au and Pd fluorescence X-rays versus the radiation pattern of the noise photons (Compton and bremsstrahlung) for detector angles between 10 and 170 degrees.

Response 2: We thank the reviewer for this suggestion. We agree with this comment and have added a panel to Figure 1 displaying both the fluorescence and background photon counts at angles between 10 and 170 degrees. We have subsequently adjusted the figure caption to “Au fluorescence significance and photon counts as simulated for an agent concentration of 1mg/ml. (a) Significance values for both Kα and Kβ plotted for each of the 9 detector angles. (b) Box chart displaying the detected photon counts for both Kα and Kβ fluorescence photons and background photons at the different detector angles.”

Comment 3: Line 438. It is intuitive understandable that dilution series (1, 438 0.33, 0.1, 0.033, 0.01, 0.0033) mg/ml are set inside the simulated tumor volume. But it is not stated on paper.

Response 3: We agree and have made the change. The new sentence reads as follows:
A dilution series with an agent concentration of [1, 0.33, 0.1, 0.033, 0.01, 0.0033] mg/ml inside the simulated tumor volume was performed in each scenario to cover a broad range of concentrations.” (Page 12, line 467-469 of the revised manuscript)

Comment 4: At line 444, Relatively to what is the X-ray beam horizontally polarized?

Response 4: We apologize for the missing clarification and have edited the respective sentence. We have changed the statement “A monoenergetic, horizontally polarized photon X-ray beam of 0.5 mm radius containing a total photon number of 1010 was used to keep the applied dose low while maintaining sufficient signal intensity.” to “A monoenergetic photon X-ray beam of 0.5 mm radius, horizontally polarized relative to the lab frame and containing a total photon number of 1010 was used to keep the applied dose low while maintaining sufficient signal intensity.” (Page 12, line 474-476)

We thank the reviewer again for taking the time to assess our manuscript and would be pleased to respond to any further questions and comments.

References:

Ito M, Shibamoto Y, Ayakawa S, Tomita N, Sugie C, Ogino H. Low-dose whole-body irradiation induced radioadaptive response in C57BL/6 mice. J Radiat Res. 2007 Nov;48(6):455-60. doi: 10.1269/jrr.07022. Epub 2007 Aug 31. PMID: 17785936.

Parkins CS, Fowler JF, Maughan RL, Roper MJ. Repair in mouse lung for up to 20 fractions of X rays or neutrons. Br J Radiol. 1985 Mar;58(687):225-41. doi: 10.1259/0007-1285-58-687-225. PMID: 4063664.

Sharma NK, Holmes-Hampton GP, Kumar VP, Biswas S, Wuddie K, Stone S, Aschenake Z, Wilkins WL, Fam CM, Cox GN, Ghosh SP. Delayed effects of acute whole body lethal radiation exposure in mice pre-treated with BBT-059. Sci Rep. 2020 Apr 22;10(1):6825. doi: 10.1038/s41598-020-63818-7. PMID: 32321983; PMCID: PMC7176697.